# Mapping SBiP1 protein-protein interactions in *Symbiodinium microadriaticum* CassKB8 using the Yeast Two-Hybrid assay and structural prediction

Estefanía Morales-Ruiz[¤], Tania Islas-Flores, Marco A. Villanueva [ID]*

Unidad Académica de Sistemas Arrecifales, Instituto de Ciencias del Mar y Limnología, Universidad Nacional Autónoma de México-UNAM, Prol. Avenida Niños Héroes S/N, Puerto Morelos, Quintana Roo, México

¤ Current Address: Universidad Politécnica de Quintana Roo, Avenida Arco Bicentenario, Mza. 11, Lote 1119-33, Sm 255, Cancún, Quintana Roo 77500, México
* marco@cmarl.unam.mx

## Abstract

BiP chaperones are central regulators of protein folding, endoplasmic reticulum homeostasis, and stress response across eukaryotes, but their client networks remain largely uncharacterized in Symbiodiniaceae. Here, we performed an initial yeast two-hybrid screening using a spliced-leader-based cDNA library to explore the interactome and regulatory dynamics of the Symbiodiniaceae BiP homolog, SBiP1, from *Symbiodinium microadriaticum* CassKB8. Using this approach, we identified eight candidate interactors with functions in translation, redox balance, RNA processing, and photosynthesis. AlphaFold2 structural modeling and Foldseek similarity analysis supported the plausibility of these candidates as BiP clients, revealing shared structural features such as globular folds and exposed hydrophobic or basic surfaces. Notably, two of the eight candidates, POX18 and TARBP1 were recovered in multiple independent clones, and even unannotated candidates displayed BiP-compatible surfaces. Gene expression analysis by RT-qPCR revealed dynamic transcriptional regulation of *SBiP1*, *HSP70*, and *POX18* over an 18-day growth time course. All three genes peaked between days 6 and 12, suggesting coordination with intrinsic cellular cycles. *SBiP1* expression was not significantly affected by light availability while *HSP70* showed a modest but statistically significant increase following cycloheximide treatment, indicating transcript stabilization. In contrast, *SBiP1* and *POX18* expressions under the latter condition remained stable, suggesting their regulation may occur post-transcriptionally. Together, our findings highlight SBiP1 as a central player in ER proteostasis, bridging multiple cellular pathways essential for stress resilience. This work provides the first structural and functional map of a BiP-centered interaction network in a photosynthetic dinoflagellate, contributing to our molecular understanding of stress adaptation in these microorganisms.

**Data availability statement:** All relevant data are within the paper and its Supporting Information file.

**Funding:** Grant 285802 to MAV, and post-doctoral fellowship to EMR from SECIHTI (Secretary of Science Humanities and Technology). The Institute of Ocean Sciences and Limnology (ICML-UNAM) provided the APC. The funders had no role in study design, data collection and analysis, decision to publish, or preparation of the manuscript.

**Competing interests:** The authors have declared that no competing interests exist.

## Introduction

The Symbiodiniaceae family encompasses various genera of photosynthetic dinoflagellates that establish symbiotic relationships with cnidarians including reef-forming corals. These microorganisms play a critical role in coral physiology and reproduction as they provide essential nutrients through photosynthesis [1–3]. However, Symbiodiniaceae are highly sensitive to environmental stressors, including extreme changes in ambient temperature, light intensity fluctuations, and nutrient availability, all of which can compromise cellular homeostasis and lead to disbyosis and coral bleaching [4,5]. On the other hand, some Symbiodiniaceae species can develop thermal tolerance after repeated exposure to heat stress [6–8].

Symbiodiniaceae have evolved several stress-response mechanisms-likely to cope with adverse conditions-such as alterations in lipid composition [9], differential production of reactive oxygen species [10], and the induction of molecular chaperones, including heat shock proteins (HSPs) [8,11,12]. Interestingly, previous transcriptomics studies using quantitative real-time PCR have shown that gene expression changes in Symbiodiniaceae often occur at relatively low fold changes. This has led to the hypothesis that regulatory processes may rely more heavily on translational or post-translational mechanisms [3,11,13,14]. Such regulation may include constitutive expression of key stress-related genes, whose functional relevance is maintained at the protein level rather than through strong transcriptional modulation [3].

In photosynthetic organisms such as *Arabidopsis thaliana* and *Chlamydomonas reinhardtii*, BiP proteins (Binding Immunoglobulin Proteins), members of the Hsp70 family play essential roles in protein folding, endoplasmic reticulum (ER) quality control, and stress signaling [15]. In *Arabidopsis*, BiP is required for normal seed development, unfolded protein response (UPR) activation, and tolerance to ER stress [16,17]. In *Chlamydomonas*, chloroplast-localized Hsp70 chaperones assist in thylakoid membrane biogenesis and the assembly of photosynthetic complexes such as VIPP1 [18]. These findings suggest that BiP chaperones in photosynthetic lineages support both general homeostasis and photosynthesis-related functions.

BiP cellular functions depend on transient interactions with a broad array of client proteins which include nascent polypeptides, misfolded proteins, and co-chaperones [19]. It also has an essential role assisting in the folding of nascent polypeptides and in the correct refolding or degradation of misfolded proteins [20]. In model systems, BiP has been shown to interact with proteins involved in lipid metabolism, oxidative stress response, and vesicular transport [21]. These interactions maintain ER homeostasis and modulate signaling cascades linking environmental stress to transcriptional reprogramming. Therefore, mapping the BiP interactome is fundamental to understand its coordination with cellular responses beyond protein folding.

BiP proteins in the ER play a central role in protein folding, quality control, and stress response [21–23]. In *Symbiodinium microadriaticum*, the BiP homolog SBiP1 is hypothesized to perform similar roles under stress, but its interaction network remains largely unexplored [24,25].

BiP recognizes its client proteins primarily through exposed hydrophobic segments and unstructured regions, typically in nascent or misfolded conformations [26,27]. Binding and release of clients are regulated by the BiP ATPase cycle and co-chaperones such as J-domain proteins (JDPs), which transiently stabilize interactions with substrate proteins [20]. These characteristics allow BiP to interact with a wide range of substrates with diverse functions. Structural features of clients, such as linear hydrophobic motifs or flexible loops, often determine their affinity for BiP. Therefore, predicting the 3D structures of candidate interactors and analyzing their surface properties can provide insights into their potential as authentic BiP clients.

While BiP has been extensively studied in model organisms, its specific role in Symbiodiniaceae remains poorly characterized. Given the central importance of chaperone-mediated stress responses in maintaining the stability of the coral-dinoflagellate symbiosis, elucidating the interaction network of SBiP1 could reveal the metabolic and cellular pathways it modulates, especially under stress conditions. In this context, identifying the client proteins of SBiP1 is a key step toward understanding its functional scope.

The yeast two-hybrid (Y2H) system is a widely used method to detect binary protein-protein interactions *in vivo* [28] and is especially valuable for studying organisms like those of the Symbiodiniaceae family which lack advanced genetic manipulation tools. Previous work has demonstrated the utility of Y2H in dinoflagellates and other algae for identifying functional networks despite their large and complex genomes [29,30]. Additionally, we used a previously published DINO-SL-based cDNA library generated from cultures of *Symbiodinium microadriaticum* CassKB8 (from now on also referred to as CassKB8), which increases the likelihood of recovering physiologically relevant interactors [29].

Based on BiPs conserved functions in ER homeostasis and stress signaling, we hypothesized that SBiP1 must interact with client proteins involved in photosynthesis, oxidative stress response, and protein trafficking. We aimed to perform an initial Y2H screening to identify and characterize putative SBiP1 interactors, providing a first view of its potential client network and infer their roles in environmental adaptation.

## Materials and methods

### CassKB8 cell cultures

Cultures of the photosynthetic dinoflagellate *Symbiodinium microadriaticum* CassKB8 (also known as MAC-CassKB8 and previously classified as clade A *Symbiodinium*) originally isolated from the jellyfish *Cassiopea xamachana,* were a kind gift of Dr. Mary Alice Coffroth (State University of New York at Buffalo). Cultures were maintained *in vitro* in ASP-8A medium under a 12 h light (80−120 μmole photon $m^{-2}$ $s^{-1}$)/12 h dark photoperiod at 25°C. Yeast strains AH109 and Y187 were grown on YPDA agar plates at 30°C. *Escherichia coli* cells were grown in LB medium at 37°C.

### RNA extraction and cDNA synthesis

Total RNA was extracted from 15-day-old CassKB8 cultures ($1 \times 10^7$ cells/mL) using the TRI Reagent (Sigma, St. Louis, MO, USA). Cells were harvested by centrifugation at $2,600 \times g$ for 5 min at 4°C, resuspended in 1 mL TRI Reagent, and disrupted with 425–600 μm glass beads in a bead beater (Biospec Products, Inc., Bartlesville, OK, USA) using four cycles of 30 s shaking and 30 s cooling. Following chloroform extraction and phase separation, RNA was precipitated with 0.1 volumes of 3 M sodium acetate at –20°C for 1 h, washed three times with 70% ethanol, and eluted in TE buffer. RNA was further purified using the RNeasy Kit (Qiagen, Hilden, Germany) and treated with RNase-free DNase I (Thermo Fisher Scientific, Waltham, MA, USA). RNA integrity was verified by 1% agarose bleach gel electrophoresis [31], and purity and concentration were determined with a spectrophotometer (BioSpectrometer; Eppendorf, Hamburg, Germany) at 260/280 nm. cDNA synthesis was performed using the SuperScript™ III Reverse Transcriptase kit (Invitrogen, Carlsbad, CA, USA). A total of 1 μg RNA was primed with oligo(dT)$_{20}$ and reverse-transcribed following the manufacturer's protocol. Residual RNA was removed with *E. coli* RNase H (New England Biolabs, Ipswich, MA, USA).

 

## Amplification and cloning of *SBiP1*

The full coding region of *SBiP1* was amplified by PCR using DreamTaq DNA polymerase (Thermo Fisher Scientific) and gene-specific primers incorporating restriction sites. The forward primer (5'-TCCCCCGGGGATGTGGAAAGTAG CCTTTG-3') contains a *Sma*I site (underlined), and the reverse primer (5'-CGGGATCCCGTTACAGCTCGTCGTGT GCC-3') includes a *BamH*I site (underlined). PCR conditions were: 95°C for 1 min; 30 cycles of 95°C for 30 s, 56°C for 30 s, and 72°C for 2 min; followed by a final extension at 72°C for 10 min. The PCR product and pGBKT7 vector were digested with *Sma*I and *BamH*I, ligated using T4 DNA ligase (Thermo Fisher Scientific), and the resulting construct pGBKT7-*SBiP1* was verified by sequencing.

## Yeast transformation and library screening

Yeast transformations were performed using the Yeastmaker™ Transformation System 2 (Clontech, Mountain View, CA, USA) following the manufacturer's small-scale protocol. The bait plasmid pGBKT7-SBiP1 was introduced into the AH109 strain, and a pre-constructed CassKB8 DINO-SL-based cDNA library fused to the GAL4 activation domain was hosted in the Y187 strain. To ensure the suitability of SBiP1 as bait, we tested for potential toxicity, auto-activation, and expression in yeast cells. Competent cells transformed with either the empty vector pGBKT7 (control) or the bait construct pGBKT7-*SBiP1* grew normally on -Trp and -Ade plates and remained white on X-α-gal, confirming lack of toxicity and absence of auto-activation. No growth was observed on -His plates, further confirming that *SBiP1* alone did not activate reporter genes. In addition, western blot analysis of total protein extracts from AH109 and Y187 strains using an anti-c-Myc antibody detected a band of the expected molecular weight only in cells carrying pGBKT7-*SBiP1*, confirming expression of the bait protein in both yeast strains. Yeast mating was then conducted as described in the Matchmaker™ Library Construction & Screening Kit manual (Clontech). After 3 weeks of incubation at 30°C in the dark, 20 colonies grew on quadruple-dropout (QDO) selection plates. Colonies were streaked three times to allow plasmid segregation, and 11 of them consistently maintained growth on selective medium.

## Plasmid recovery and sequencing

Plasmids were recovered from positive yeast clones using the Zymoprep Yeast Plasmid Miniprep II kit (Zymo Research, Irvine, CA, USA), then transformed into chemically competent *E. coli* DH5α cells via the $CaCl_2$ heat shock method. Plasmids were extracted from bacterial cultures using the GeneJET Plasmid Miniprep Kit (Thermo Fisher Scientific) and sent to a sequencing facility (see below) to identify putative SBiP1-interacting proteins. Nucleotide sequences were trimmed and assembled, and each insert was analyzed using BLAST against the NCBI database to identify the most probable encoded protein. Eight potential ligands out of 11 positive clones were identified. The resulting top-scoring matches were used to infer candidate protein identity, functional category, and to guide subsequent structural predictions. Sequences of all positive clones were compiled into a FASTA file (S1 Appendix). All sequencing was carried out as external service at the facility of the Instituto de Biotecnología of UNAM in Cuernavaca, Morelos, México.

## 3D modelling and structural analysis

The predicted amino acid sequences of the SBiP1-interacting proteins identified by Y2H screening were modeled using AlphaFold2 via the ColabFold notebook [32]. Sequences were input in FASTA format, and models were generated with default parameters. The structural confidence of the predictions was evaluated based on the predicted Local Distance Difference Test (pLDDT) scores. Resulting models were visualized and analyzed using UCSF ChimeraX [33] to examine putative domains and interaction surfaces. Structural comparisons were performed when appropriate to assess potential functional motifs or domain conservation. Structural similarity was evaluated using Foldseek v5.0 [34], which performs ultra-fast comparisons of 3D protein models against the Protein Data Bank (PDB). AlphaFold-predicted structures were

uploaded to the Foldseek webserver (https://search.foldseek.com) and searched against the AlphaFoldDB or PDB structures using default parameters. The top hit for each model was selected and reported.

### Gene expression analysis by quantitative real-time PCR and statistical analysis

To assess the expression dynamics of *SBiP1* and its potential interacting partners, *HSP70* and *POX18*, quantitative real-time PCR (qPCR) was performed under different experimental conditions. A batch culture of CassKB8 was grown in ASP-8A medium at 25°C under a 12:12 h light/dark photoperiod. Samples were collected at multiple time points and treatments to examine basal expression, light-dependence, and cycloheximide (CHX) response. For basal expression analysis, samples were collected on days 3, 6, 9, 12, and 15, six hours after light onset. For the light/dark experiment, samples were taken on days 3, 12, and 18 from cultures maintained under either light or dark conditions. For the CHX treatment, cultures were exposed on day 12–100 µg/mL CHX or a vehicle control and harvested six hours later. Each treatment or time point consisted of three biological replicates, each with three technical replicates. Total RNA was extracted using TRI Reagent (Sigma-Aldrich) followed by purification with the RNeasy Mini Kit (Qiagen), as described previously. First-strand cDNA synthesis was performed using 1 µg of total RNA and SuperScript™ III Reverse Transcriptase (Invitrogen) with a combination of random hexamers and oligo(dT) primers. The integrity of the cDNA was confirmed using primers designed to amplify the full open reading frame of *SmicRACK1*. qPCR was performed using a StepOnePlus™ Real-Time PCR System (Applied Biosystems, Waltham, MA, USA) and the QuantiTect SYBR Green PCR Kit (Qiagen), in 20 µL reactions with three technical replicates per biological sample. *Elongation factor 1-α* from *S. microadriaticum* CassKB8 (*SmicEF1-α*) was used as the reference gene for normalization as it has been previously validated for stability in qPCR assays in numerous organisms and physiological stages [35–42], including microalgae [39,40] and Symbiodiniaceae in culture [41,42]. Relative expression was calculated using the $2^{-\Delta\Delta Ct}$ method. For all experiments, the control or baseline group (e.g., day 3 light for temporal and diel studies, or untreated control for CHX assays) was used as the calibrator. For time-course experiments, one-way ANOVA followed by Tukey's Honest Significant Difference (HSD) post-hoc test was used to detect differences between time points. For light/dark data, a two-way ANOVA was performed to assess the effects of day, light condition, and their interaction. In the CHX treatment experiments, Welch's t-tests were used to compare expression between treated and control groups for each gene. A significance threshold of $\alpha = 0.05$ was applied. All statistical analyses were performed in RStudio (v2023.06.1 + 524) using R (v4.2.1). The following functions and packages were used: one-way ANOVA with aov(), Tukey's HSD with TukeyHSD(), Welch's t-test with t.test(..., var.equal = FALSE), and non-parametric tests with kruskal.test(). Significance letters were generated using the multcompView package, and bar plots were produced with ggplot2 using stat_summary() for error bars.

## Results

### Identification and annotation of candidate interactors

To explore the protein-protein interaction landscape of SBiP1, a Y2H screen was performed using a CassKB8 DINO-SL-based cDNA library as prey. A total of 11 positive clones were recovered after stringent selection, each representing a potential SBiP1-interacting protein. Plasmids were extracted from yeast, transformed into *E. coli*, and re-isolated for sequencing of the cDNA inserts. To identify the proteins encoded by the eleven sequenced cDNA inserts recovered from the Y2H screen, each nucleotide sequence was analyzed using BLAST against the NCBI database. The top-scoring hit for each candidate was selected based on total alignment score and used to infer protein identity and potential function. These annotations served as the basis for initial classification of interactors prior to structural modeling. A summary of the best BLAST hits, including protein description, source organism, E-value, sequence identity, coverage, and accession numbers, are provided in S1 Table.

## Structural predictions and fold-level functional inference of SBiP1 interactors

To better understand the structural and functional properties of the SBiP1-interacting proteins identified in the Y2H screen, we predicted their three-dimensional conformations using AlphaFold2 and compared them against experimentally resolved structures using Foldseek [34]. The corresponding amino acid sequences were submitted to ColabFold, and resulting models were evaluated based on per-residue confidence scores (pLDDT) and overall fold quality (Fig 1 and S2 Table).

Most interactors displayed well-folded, compact structures with high-confidence predictions (mean pLDDT > 85), particularly in regions corresponding to annotated functional domains. To facilitate domain-level functional inference, each predicted model was aligned to structural databases using Foldseek, allowing comparison to experimentally resolved proteins and refinement of functional annotations. Several candidates showed clear structural similarity to well-characterized proteins, reinforcing and refining their predicted functions. The HSP70 candidate (C1) exhibited a canonical two-lobed fold with a defined ATPase domain, consistent with DnaK/HSP70 family chaperones. The peroxisomal protein POX18 (C2/5/10) matched an enoyl-CoA hydratase domain involved in fatty acid β-oxidation. The elongation factor-like protein EFL (C9) aligned well with EF1-α, while the putative RNA methyltransferase TARBP1 (C4/11) showed strong similarity to nucleolar protein 58 (NOP58), suggesting a potential role in snoRNP assembly rather than methylation. Likewise, MAP2B (C6) and GTPBP1 (C7) shared folds with ribosomal proteins L29 and eL24, respectively, supporting roles in

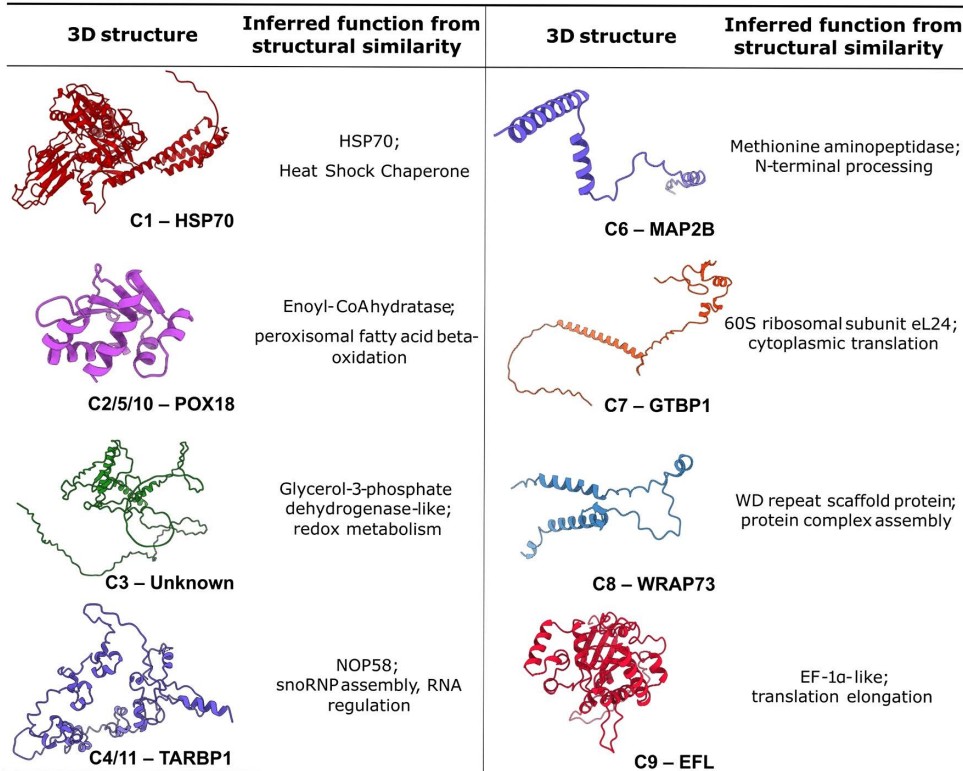

| 3D structure | Inferred function from structural similarity | 3D structure | Inferred function from structural similarity |
|---|---|---|---|
| C1 – HSP70 | HSP70; Heat Shock Chaperone | C6 – MAP2B | Methionine aminopeptidase; N-terminal processing |
| C2/5/10 – POX18 | Enoyl-CoA hydratase; peroxisomal fatty acid beta-oxidation | C7 – GTBP1 | 60S ribosomal subunit eL24; cytoplasmic translation |
| C3 – Unknown | Glycerol-3-phosphate dehydrogenase-like; redox metabolism | C8 – WRAP73 | WD repeat scaffold protein; protein complex assembly |
| C4/11 – TARBP1 | NOP58; snoRNP assembly, RNA regulation | C9 – EFL | EF-1α-like; translation elongation |

**Fig 1. Predicted 3D structures and inferred functions of SBiP1-interacting proteins.** Three-dimensional models of candidate SBiP1-interacting proteins identified by yeast two-hybrid screening. Predictions were generated using AlphaFold2 (ColabFold implementation) and assessed by per-residue confidence scores (pLDDT). Structural similarity searches were performed using Foldseek to infer putative functions. Models include HSP70 (C1), POX18 (C2/5/10), an unannotated globular α-helical protein (C3), TARBP1 (C4/11) with structural similarity to NOP58, MAP2B (C6), GTPBP1 (C7), WRAP73 (C8), and EFL (C9).

cytoplasmic translation. Notably, the previously unannotated candidate (C3) yielded a coherent, globular α-helical model despite lacking sequence homology. Foldseek analysis identified significant structural similarity to a glycerol-3-phosphate dehydrogenase-like protein, suggesting a possible role in redox metabolism.

Based on predicted annotations and Gene Ontology terms, these interactors can be grouped into functional categories that align with known roles of BiP in cellular homeostasis as follows: (1) protein folding and quality control (HSP70); (2) redox and metabolic processes (POX18, C3); (3) RNA binding and modification (TARBP1); (4) protein synthesis and processing (EFL, methionine aminopeptidase); and (5) cytoskeletal or scaffolding functions (WRAP73). These functions are consistent with BiP proteins involvement in assisting protein maturation, mitigating stress-induced damage, and facilitating the assembly of macromolecular complexes.

Finally, inspection of model surfaces revealed several candidates with hydrophobic or basic patches consistent with BiP client-binding sites. These features may facilitate transient recognition by SBiP1, which preferentially associates with unfolded or partially folded substrates enriched in non-polar residues. A summary of all candidate interactors, including their annotation, predicted function, estimated molecular weight, fold-level similarity, and functional classification, is presented in Table 1.

## Basal expression dynamics of *SBiP1*

To determine whether *SBiP1* is transcriptionally regulated under standard growth conditions, we analyzed its basal expression profile over a 15-day growth period. Given its central role in protein folding and stress responses, understanding baseline regulation of *SBiP1* is key to interpreting its potential interactions and downstream effects. Relative transcript levels were quantified by qPCR and normalized to the reference gene *SmicEF1-α* [35–42]. Fold changes were calculated using the $2^{-\Delta\Delta Ct}$ method, with day 3 serving as the calibrator. A one-way ANOVA revealed a significant effect of time on *SBiP1* expression ($p < 0.05$). Post-hoc Tukey HSD tests indicated that expression peaked at day 12 (fold change = $2.62 \pm 0.08$) and was significantly higher than at all other time points. Expression at day 6 ($1.77 \pm 0.12$) was also elevated and statistically

**Table 1. Summary of SBiP1-interacting proteins identified by Yeast Two-Hybrid screening.**

| Candidate | Annotation | Predicted function | Molecular weight (kDa) | Functional category | Structural domains | Predicted interaction surface |
|---|---|---|---|---|---|---|
| 1 | HSP70 | Protein folding | 80.38 | Chaperone | Canonical ATPase and SBD domains | Hydrophobic cleft (substrate-binding domain) |
| 2, 5, 10 | POX18 | Beta-oxidation | 48.27 | Oxidative stress, metabolism | Multidomain: enoyl-CoA hydratase + HAD | Hydrophobic patches near HAD-like domain |
| 3 | Unknown | — | 21 | — | Alpha-helical, globular | Basic and nonpolar residues exposed |
| 4, 11 | TARBP1 | RNA methylation | 120 | RNA metabolism | SAM-binding methyltransferase fold | Flexible loops around SAM pocket |
| 6 | MAP2B | Methionine processing | 52.8 | Protein processing | Compact M24 peptidase-like | Shallow cavity in C-terminal domain |
| 7 | GTPBP1 | mRNA decay modulation | 86.6 | Translation regulation | GTPase fold, alpha/beta | Polar loop adjacent to GTP pocket |
| 8 | WRAP73 | Scaffold | 125.3 | Cytoskeletal, protein binding | Beta-propeller WD40 repeats | Distributed surface, scaffold-like |
| 9 | EFL | Elongation factor | 52.45 | Translation | EF-Tu GTPase-like domain | Hydrophobic loop near GTP site |

Candidate proteins identified as putative interactors of the ER chaperone SBiP1 in *Symbiodinium microadriaticum* CassKB8. Functional annotations were derived from BLAST results, Gene Ontology terms, and fold-level similarity analyses performed with Foldseek. **Abbreviations:** HAD, hydroxyacyl-CoA dehydrogenase; SAM, S-adenosylmethionine; SBD, substrate-binding domain.

distinct from the remaining days. No significant differences were found among days 3, 9, and 15, all of which displayed similar basal levels near or below the calibrator (Fig 2). These results indicate a *SBiP1* basal expression along the growth cycle except at two specific time points which may be physiologically important during cell growth.

Given the diel oscillations in dinoflagellate metabolism and the central role of BiP in protein folding, plus the previous findings on dark/light-modulated phosphorylation [24,25,43,44], we assessed whether *SBiP1* transcript levels were also modulated by light conditions. Samples were collected at days 3, 12, and 18, six hours after the onset of either light or dark phases. Transcript levels were normalized to *SmicEF1-α* and calculated using the $2^{-\Delta\Delta Ct}$ method, with day 3 under light conditions as the calibrator. A two-way ANOVA revealed a significant effect of time on *SBiP1* expression ($p < 0.05$), but neither light condition ($p = 0.82$) nor its interaction with time ($p = 0.54$) showed statistical significance. Tukey HSD post-hoc tests confirmed significantly elevated expression at day 12 under both conditions (Light: 1.52 ± 0.03; Dark: 1.56 ± 0.15), whereas at days 3 and 18, it maintained similarly low levels (~ 1.0; Fig 3). These results suggest that *SBiP1* expression is modulated over time but not directly driven by light availability.

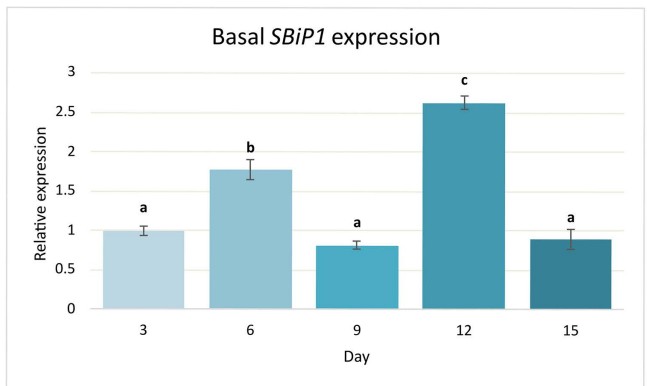

**Fig 2. Basal expression of *SBiP1* during a time course under standard culture conditions.** Relative transcript levels of *SBiP1* were measured at five time points (days 3, 6, 9, 12, 15) under standard culture conditions. Values were normalized to *SmicEF1-α* and calculated using the $2^{-\Delta\Delta Ct}$ method with day 3 as calibrator. Bars represent mean ± SE (n = 9 per time point). Different letters above bars indicate statistically significant differences (one-way ANOVA followed by Tukey's HSD, α = 0.05).

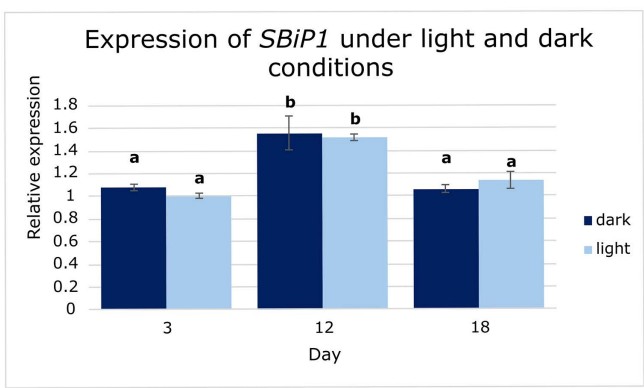

**Fig 3. Expression of *SBiP1* under light and dark conditions.** Transcript levels of *SBiP1* at days 3, 12, and 18 in cultures maintained under light or dark conditions. Values were normalized to *SmicEF1-α* and expressed relative to day 3 (light). Bars represent mean ± SE (n = 9, from 3 biological replicates × 3 technical replicates treated as independent values). Different letters indicate significant differences between groups (two-way ANOVA, Tukey's HSD, α = 0.05).

## Basal and condition-dependent gene expression patterns of candidate genes

Among the interactors identified by the Y2H screen, *HSP70* and *POX18* transcripts were selected for further analysis due to their known involvement in stress adaptation. Before proceeding with expression analyses, their interactions with SBiP1 were reconfirmed by direct yeast mating assays using freshly transformed bait (ligand) and prey (SBiP1) yeast strains, ensuring that these candidates represented true positives from the initial screening. *HSP70* is a canonical molecular chaperone frequently co-regulated with BiP, while *POX18* is involved in peroxisomal β-oxidation and may contribute to redox balance and membrane remodeling. To assess the baseline transcriptional activity of these SBiP1-interacting proteins, we analyzed the expression dynamics of *HSP70* and *POX18* under standard and experimental conditions using qPCR. Each condition included three biological replicates, each with three technical replicates. Transcript levels were normalized to the reference gene *SmicEF1-α*, and relative expression was calculated using the $2^{-\Delta\Delta Ct}$ method with appropriate calibrator groups. Each time point included three biological replicates with three technical replicates each. Expression of *POX18* was quantified over an 18-day growth time course. Given POX18 role in β-oxidation and potential involvement in redox homeostasis, we hypothesized that it could display dynamic basal expression even in the absence of stress. One-way ANOVA revealed a significant effect of time on its expression ($p < 0.05$). Tukey HSD post-hoc tests showed that expression at days 12 (3.55 ± 0.25-fold) and 15 (4.21 ± 0.26-fold) was significantly higher than at day 3 (1.00 ± 0.04-fold), while at day 18 it returned to baseline levels (Fig 4). These results suggest moderate transcriptional regulation of *POX18* even in the absence of acute stress.

Similarly, *HSP70*, which encodes a canonical molecular chaperone involved in protein quality control, exhibited significant temporal modulation ($p < 0.05$). Tukey HSD post-hoc analysis revealed that expression increased sharply by day 6 (6.48 ± 0.54-fold), remained elevated at days 9 (5.05 ± 0.09-fold) and 12 (6.32 ± 0.25-fold), but stayed above basal levels at days 15 and 18 (≈ 4.0-fold; $p < 0.001$ vs. day 3) (Fig 5). The relatively pronounced transcript-level changes that we observed are larger than those previously reported for Symbiodiniaceae genes [3,11,14], and suggest that both *POX18* and *HSP70* may be transcriptionally regulated in coordination with intrinsic metabolic or developmental cycles. Nonetheless, post-translational mechanisms to fine-tune the activity of the expressed proteins are also likely to occur, as observed for many stress-related proteins in dinoflagellates [3,11,13,14].

## Effect of CHX on *SBiP1* and client gene expression

Since we have previously shown that CHX inhibits the dephosphorylation of SBiP1 and that this inhibition is abrogated by heat shock [44], we sought to explore the effect of CHX on the transcript levels of *SBiP1* and its two putative clients

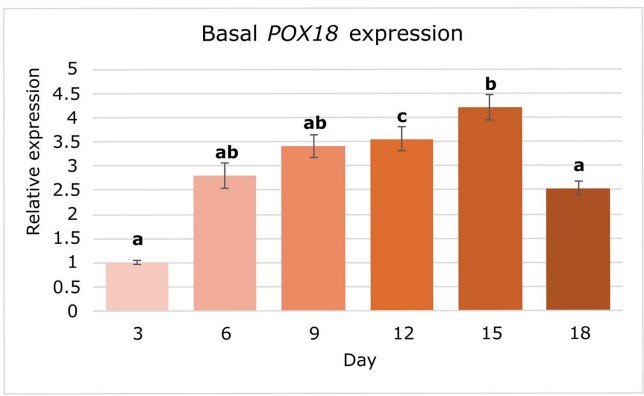

**Fig 4. Basal expression profile of *POX18* during a time course under standard conditions.** Relative transcript levels of *POX18* across six time points, normalized to *SmicEF1-α* and expressed relative to day 3. Bars represent mean ± SE (n = 9 per time point). Different letters above bars indicate significant differences (one-way ANOVA followed by Tukey's HSD, α = 0.05).

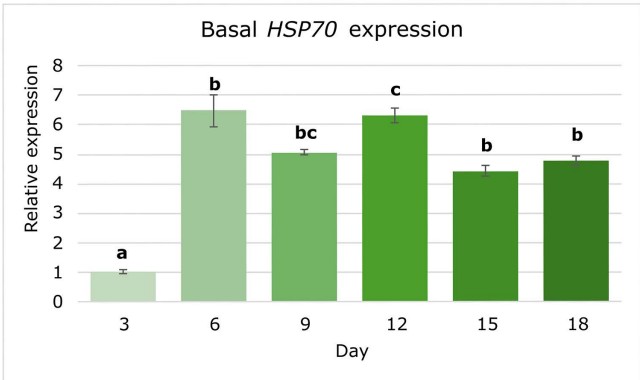

**Fig 5. Basal expression profile of *HSP70* during a time course under standard conditions.** Relative transcript levels of *HSP70* across six time points, normalized to *SmicEF1-α* and expressed relative to day 3. Bars represent mean±SE (n=9 per time point). Different letters above bars indicate significant differences (one-way ANOVA followed by Tukey's HSD, α=0.05).

*HSP70* and *POX18*, to examine whether their expression was linked to protein biosynthesis. When the cultures were treated with CHX on day 12, a Welch's t-test revealed a small but statistically significant increase in *HSP70* expression ($p=0.036$) compared to control (Fig 6, orange bar, *HSP70*), while *SBiP1* and *POX18* showed the same upward trend without reaching statistical significance ($p=0.168$ and $p=0.132$, respectively), (Fig 6, orange bars, *POX18* and *SBiP1*, respectively). These results suggest that translation inhibition may lead to a general, though modest, increase in transcript accumulation, rather than a gene-specific feedback stabilization effect.

## Discussion

### Structural insights into SBiP1 interactions

The identification of SBiP1-interacting proteins in CassKB8 reveals new insights into the molecular roles of ER-resident BiP chaperones in Symbiodiniaceae and their potential contribution to environmental stress tolerance. The diversity of recovered interactors suggests that SBiP1 assists in the folding and stabilization of client proteins involved in photosynthesis, protein processing, translation, RNA metabolism, and oxidative stress management.

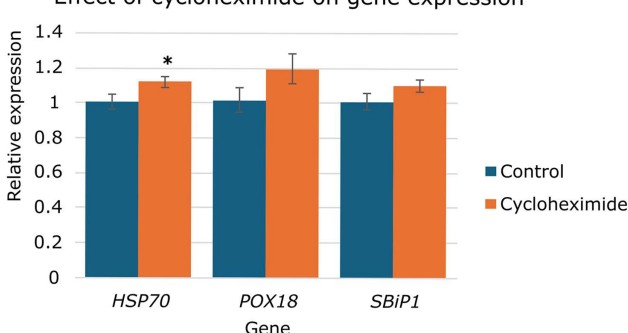

**Fig 6. Effect of cycloheximide on the expression of *SBiP1*, *HSP70*, and *POX18*.** Transcript levels of *SBiP1*, *HSP70*, and *POX18* at day 12 in control and CHX-treated cultures. Expression values were normalized to *SmicEF1-α* and expressed relative to untreated controls. Bars represent mean±SE (n=3 biological replicates, with technical triplicates averaged within each biological). The asterisk indicates significant difference between treated and control groups ($p<0.05$, Welch's t-test).

The interaction with photosynthesis-related proteins such as elongation factors and chaperones points to a potential role for SBiP1 in supporting chloroplast function and protein synthesis during light or thermal stress [4,45]. These stressors are known to destabilize the photosynthetic machinery and are implicated in coral bleaching events [46]. The identification of POX18, a peroxisomal multifunctional enzyme involved in fatty acid β-oxidation [47] as a putative SBiP1 ligand, suggests a possible link between ER homeostasis and redox balance. Such a connection may be critical for symbiosis under heat stress, when reactive oxygen species (ROS) accumulation becomes detrimental to both the symbiont and host tissues [48,49].

Structural modeling using AlphaFold2 further supports the plausibility of these candidates as SBiP1 clients. Most interactors showed well-folded, high-confidence structures, often containing surface-exposed hydrophobic or basic patches consistent with known BiP binding preferences. Thus, several proteins exhibited well-characterized domain architectures such as ATPase and substrate-binding domains in HSP70, SAM-binding folds in TARBP1, and enoyl-CoA hydratase/dehydrogenase domains in POX18, that are typical of chaperone-assisted substrates. On the other hand, in multiple candidates including MAP2B and GTPBP1, flexible loops or shallow binding pockets were observed near predicted catalytic or ligand-interacting regions, suggesting potential SBiP1 contact surfaces (Fig 1).

Interestingly, the candidate initially annotated as TARBP1 displayed stronger structural similarity to nucleolar protein 58 (NOP58), a core component of snoRNP complexes, than to canonical RNA methyltransferases [50]. This suggests that the protein was misannotated and could function in RNA processing and snoRNP assembly rather than in methylation. Such cases underscore the importance of integrating structure-based predictions with sequence homology, particularly in Symbiodiniaceae where annotations based solely on BLAST can be misleading. This has been precisely the case for SBiP1 whose annotated sequences had to be corrected twice after careful analysis [24,25]. Notably, *POX18* and *TARBP1* were independently recovered multiple times in the screen, further supporting their physiological relevance. Even the candidate (C3) yielded a globular α-helical fold with exposed basic and hydrophobic residues, indicating structural compatibility with BiP binding despite the lack of functional annotation. This highlights the capacity of structure-based comparisons to uncover functional clues in poorly characterized or novel sequences, a particularly valuable approach in Symbiodiniaceae where sequence homology is often limited. While the interactors span a range of biological functions, they share common structural features-such as globular architectures and exposed, flexible motifs-that support their potential recognition by a generalist ER chaperone like SBiP1 (Table 1).

## Expression dynamics of SBiP1 and client genes

To determine whether *SBiP1* transcript levels vary over time under standard culture conditions, we analyzed its expression across a five-point growth time-course. One-way ANOVA revealed a highly significant temporal effect. Expression increased at day 6 and peaked sharply at day 12, significantly higher than all other time points (Fig 2). This suggests transcription processes in coordination with intrinsic metabolic or developmental cycles rather than in direct response to light. This would also be consistent with a scenario where SBiP1 undergoes dynamic basal regulation even under non-stressful conditions, potentially anticipating cyclic cellular demands in the absence of overt stress.

Two-way ANOVA indicated that the diel variation observed in *SBiP1* expression is primarily time-dependent, with similar profiles under both light and dark conditions (Fig 3). This supports the hypothesis that transcriptional regulation of BiP is uncoupled from immediate light cues, and rather follows intrinsic metabolic or developmental cycles which has also been observed in other photosynthetic organisms under diel regulation [51,52]. In addition, previous biochemical work has shown that SBiP1 activity is modulated by reversible phosphorylation in response to light and temperature cues [24,25,43,44], supporting a likely regulation at the post-translational level through phosphorylation. Interestingly, all time points analyzed under dark/light conditions showed no variation in expression that could be light-dependent (Fig 3). These results are consistent with our previous data where no variation was observed in the SBiP1 protein levels when cells were changed form 12 h darkness to light exposure to up to 7 h [44].

To further explore the biological relevance of selected interactors, we analyzed the basal expression profiles of *HSP70* and *POX18* across a temporal light cycle and under translational inhibition. Both genes exhibited moderate yet significant temporal modulation in transcript abundance throughout the growth cycle, with *POX18* peaking around day 15 (Fig 4) and *HSP70* around day 6 (Fig 5), which suggests some degree of transcriptional regulation synchronized with metabolic activity. The observed fold-change levels (up to 4–6 FC) were relatively high compared to previous reports of transcriptional changes in Symbiodiniaceae, where lower variations were observed [3], supporting the view that transcriptional responses in this lineage are usually modest and that significant regulation probably occurs post-translationally. However, the expression patterns we observed for *HSP70* and *POX18* suggest that, at least for these genes, a substantial transcriptional regulation component underlies the observed modulation despite likely additional post-translational regulation mechanisms contributing to fine-tune their functional control [53].

The CHX experiment revealed a modest but statistically significant increase in *HSP70* transcripts under translation inhibition, while *SBiP1* and *POX18* followed the same non-statistically significant upward trend (Fig 6). These results suggest a general effect of translational inhibition on transcript accumulation rather than gene-specific regulation.

## Functional implications for stress resilience

The diversity of recovered interactors suggests that SBiP1 coordinates multiple cellular processes beyond protein folding, including redox metabolism, RNA processing, and cytoplasmic translation. The identification of POX18 as putative ligand, which is a peroxisomal enzyme involved in fatty acid β-oxidation, points to a potential link between ER homeostasis and redox balance [47–49]. Such a connection may be critical for symbiosis under heat stress where ROS are generated and, after accumulation, threatens both the symbiont and the host. Similarly, the interaction with translation-related proteins such as EFL and ribosomal subunits suggests that SBiP1 could support protein synthesis under fluctuating environmental conditions, safeguarding photosynthesis-related proteins that are particularly vulnerable to stress [3,13,14,45].

By linking these pathways, a role of SBiP1 as a potential regulatory hub for maintaining cellular homeostasis could be initially proposed. This may also have important implications for coral-dinoflagellate symbiosis, where the ability of the symbiont to cope with environmental stress contributes to the resilience of the holobiont.

## Limitations of the Y2H approach and qPCR observations

One notable limitation of this work is inherent to the Y2H system itself, which can yield up to ~ 50% false positives [54] and is less efficient at detecting transient or cofactor-dependent interactions [55]. Accordingly, this study should be regarded as an initial screen of the SBiP1 interactome. Thus, although selected candidates were re-tested through direct mating assays to confirm interaction specificity before further analysis, confirmation will be necessary through complementary approaches such as co-immunoprecipitation, affinity purification-mass spectrometry, or proximity labeling to further validate and expand the SBiP1 network.

In addition, the data obtained on RNA expression was obtained using *SmicEF1-α* as the reference gene above any others because it has been shown to perform adequately for RNA quantitation by qPCR in numerous species [35–40], developmental stages [35–38,40,42], and culture cycle conditions including microalgae [39,40] and Symbiodiniaceae [41,42]. However, these results must be interpreted with caution provided that only this reference gene was used, and further confirmation with additional reference genes may be desirable. In any case, the reported reliability of *EF1-α* as an adequate reference gene across many species and physiological processes along with the previous results on constant SBiP1 protein expression under either dark or light conditions [44] coincident with the results from this study (Fig 3), support the validity of the observed trends in *SBiP1*, *HSP70*, and *POX18* and our interpretation of the results.

## Concluding remarks

Altogether, this work highlights SBiP1 as a central component of the cellular stress management network in Symbiodiniaceae, bridging key metabolic and regulatory pathways. Understanding how BiP coordinates folding of diverse substrates under environmental stress enhances our molecular view of Symbiodiniaceae resilience and may inform strategies to mitigate coral bleaching under climate change scenarios.

Overall, this study suggests that *SBiP1* may function as a potential regulatory hub in the ER homeostasis network coordinating cross-talk with key metabolic pathways of CassKB8 and, most likely, of Symbiodiniaceae as we have identified SBiP1 across all previously analyzed species of this family [25,43]. However, given the limitations of the Y2H approach, additional experimental confirmation will be necessary to validate these interactions and fully establish if SBiP1 plays a central role in the proteostasis network. Understanding how SBiP1 interacts and orchestrates the folding of its diverse substrates will provide crucial insight into symbiont contending stress for resilience under the challenge of climate change. Beyond its molecular significance, this study also provides a framework to connect chaperone-mediated networks with cnidarian–dinoflagellate symbiosis.

## Supporting information

**S1 Appendix. Sequences from all positive clones obtained from the Yeast-Two Hybrid screening.**
(FASTA)

**S1 Table. Top BLAST hits for each SBiP1-interacting candidate.** Candidate sequences identified by the yeast two-hybrid screen and analyzed using BLAST against the NCBI nucleotide or protein databases.
(PDF)

**S2 Table. Summary of structural quality metrics for SBiP1-interacting candidates.** The eleven SBiP1-interacting candidates predicted using AlphaFold2 are shown.
(PDF)

## Acknowledgments

We are grateful for the technical help of M.O. Edgar Escalante-Mancera, M.I. Miguel Ángel Gómez-Reali, Dr. Edén Magaña-Gallegos, M.C. Esmeralda Pérez-Cervantes, and M.T.I.A. Gustavo Villarreal-Brito.

## Author contributions

**Conceptualization:** Marco A. Villanueva.

**Data curation:** Estefanía Morales-Ruiz.

**Formal analysis:** Estefanía Morales-Ruiz, Tania Islas-Flores.

**Funding acquisition:** Marco A. Villanueva.

**Investigation:** Estefanía Morales-Ruiz, Tania Islas-Flores, Marco A. Villanueva.

**Methodology:** Estefanía Morales-Ruiz, Tania Islas-Flores.

**Project administration:** Marco A. Villanueva.

**Supervision:** Tania Islas-Flores, Marco A. Villanueva.

**Writing – original draft:** Estefanía Morales-Ruiz, Marco A. Villanueva.

**Writing – review & editing:** Estefanía Morales-Ruiz, Tania Islas-Flores, Marco A. Villanueva.

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
