## [Decision Letter · Decision Letter 0]

27 Nov 2025

Dear Dr. Villanueva,

Thank you for submitting your manuscript to PLOS ONE. After careful consideration, we feel that it has merit but does not fully meet PLOS ONE’s publication criteria as it currently stands. Therefore, we invite you to submit a revised version of the manuscript that addresses the points raised during the review process.

We look forward to receiving your revised manuscript.

Kind regards,

Katherine A. Borkovich, Ph.D.

Academic Editor

PLOS ONE

Journal Requirements:

“Grant 285802 to MAV, and postdoctoral fellowship to EMR from SECIHTI (Secretary of  Science Humanities and Technology)”

3. Please include captions for your Supporting Information files at the end of your manuscript, and update any in-text citations to match accordingly. Please see our Supporting Information guidelines for more information: http://journals.plos.org/plosone/s/supporting-information .

Reviewers' comments:

Reviewer's Responses to Questions

**Comments to the Author**

1. Is the manuscript technically sound, and do the data support the conclusions?

Reviewer #1: Partly

Reviewer #2: Yes

2. Has the statistical analysis been performed appropriately and rigorously?

Reviewer #1: Yes

Reviewer #2: Yes

3. Have the authors made all data underlying the findings in their manuscript fully available?

Reviewer #1: Yes

Reviewer #2: Yes

4. Is the manuscript presented in an intelligible fashion and written in standard English?

Reviewer #1: Yes

Reviewer #2: Yes

Reviewer #1: The authors of this paper describe their study of using the yeast two-hybrid (Y2H) assay to investigate the interactome of the Symbiodiniaceae BiP1 (Binding Immunoglobulin Protein) homologue. BiPs belong to the heat shock protein 70 (Hsp70) family that are important in a cell’s response to stress by repairing and refolding damaged proteins in the endoplasmic reticulum as well as helping to fold and construct newly synthesised proteins. As Symbiodiniaceae form symbioses with reef-building corals, uncovering the proteins and their interactors that mitigate their stress response is important in understanding their susceptibility to stressful conditions e.g., ocean warming and the persistence of this keystone symbiosis.

This is a nice paper and provides a good initial screen for candidate proteins in the SBiP1 interactome. I recommend the paper for publication following the minor corrections detailed below.

The Y2H assay is a high throughput and low cost way to investigate the interactome and a good method for organisms like Symbiodiniaceae that are notoriously hard to genetically alter. However, due to the high rate of false positives in the Y2H (up to 50%; Von Mering et al., 2002), the authors should make it clear to the reader that further work is needed to confirm the findings e.g., refer to the interacting proteins found in this study as candidates (which the authors have done in the introduction) and explicitly say that the study is an “initial screen” of the SBiP1 interactome. Furthermore, the authors could elaborate more on the weaknesses of the Y2H assay in the limitations section of the discussion. This could include the authors suggesting another method to corroborate the findings from the Y2H assay, with two/three methods investigating protein interactions increasing the accuracy of correctly identifying the interacting proteins (Von Mering et al., 2002).

Line 478-485 - Confused with this paragraph:

Figure 4 and 5 show transcripts do change in abundance over time with growth, and the results also say that both Pox18 and Hsp70 exhibit "significant temporal modulation”.

Statement made on line 481 contrasts with sentence above - do you mean “regulated through post-translational mechanisms”?

Again, the sentence beginning on line 483 seems to contrast to the patterns seen in the figures.

Line 486 - From looking at the graph for CHX, although the transcript expression is only significantly different in Hsp70 control vs CHX treatment, the other transcripts show similar patterns (higher expression after CHX treatment vs control). I don’t think from these findings you can conclude that Hsp70 is under feedback transcript stabilisation versus SBiP1 or POX18 are transcriptionally stable and not linked to active translation. Maybe if more replicates were included all the genes would show the same significance. Authors could reword to say “Hsp70 shows a weak significant increase” and mention that the other transcripts show exactly the same pattern, but it doesn’t meet significance.

Line 503 - not sure if the findings confirm that SBiP1 is a “central hub” due to Y2H assay limitations - reword as the authors need extra confirmation of protein-protein interactions before making this statement

Typo in Figure 5 - should say “Hsp70” in graph title not “Hxp70”

Line 382 - the text says that the expression of Hsp70 peaked at day 12 but figure 5 shows it peaking at day 6. The numbers in the text for change in expression on day 12 and 6 also don’t match those on the figure - looks to be around 6.5-fold for day 6 and 6.25-fold for day 12.

Reviewer #2: The manuscript is generally well written; however, there are several areas where improvements are needed. In the Abstract, acronyms should be defined before they are used. For example, ER appears without being defined anywhere in the manuscript.

Additionally, the figures could be improved by enhancing their resolution, increasing clarity of labels, and ensuring that all elements are consistently formatted and redundant info removed (e.g., parts in brackets for Figures 2-6).

Can author add limitation regarding the way how gene expression was assessed as using only one reference genes may not be sufficient. Why was this specific reference gene SmicEF1-α used? Did you consider other genes? What was validation process here?

**Do you want your identity to be public for this peer review?** For information about this choice, including consent withdrawal, please see our Privacy Policy

Reviewer #1: **Yes:**  Lucy Mae Gorman

Reviewer #2: No

---

## [Author Response · Author response to Decision Letter 1]

8 Dec 2025

We thank both reviewers for their keen observations which have significantly improved the manuscript.

Reviewer #1:

I recommend the paper for publication following the minor corrections detailed below.

Query: The Y2H assay is a high throughput and low cost way to investigate the interactome and a good method for organisms like Symbiodiniaceae that are notoriously hard to genetically alter. However, due to the high rate of false positives in the Y2H (up to 50%; Von Mering et al., 2002), the authors should make it clear to the reader that further work is needed to confirm the findings e.g., refer to the interacting proteins found in this study as candidates (which the authors have done in the introduction) and explicitly say that the study is an “initial screen” of the SBiP1 interactome. Furthermore, the authors could elaborate more on the weaknesses of the Y2H assay in the limitations section of the discussion. This could include the authors suggesting another method to corroborate the findings from the Y2H assay, with two/three methods investigating protein interactions increasing the accuracy of correctly identifying the interacting proteins (Von Mering et al., 2002).

Response: We thank the reviewer for this valuable suggestion. The manuscript was revised to explicitly describe the study as an initial yeast two-hybrid (Y2H) screen of the SBiP1 interactome. The limitations of the Y2H approach are now further considered in a new section added to the discussion (lines 526-544) “Limitations of the Y2H approach and qPCR observations”, emphasizing the system’s high false-positive rate and the need for complementary validation methods for this approach such as co-immunoprecipitation, affinity purification–mass spectrometry, and proximity labeling (lines 526-533) as well as for the qPCR (lines 534-544). Additionally, a sentence in the Abstract was placed (lines 19-20) to read “we performed an initial yeast two-hybrid screening…”, and in the Introduction (lines 112-114) to read “We aimed to perform an initial Y2H screening to identify putative SBiP1 interactors…”. Finally, we clarified in the results (lines 362-365) that, prior to conducting gene expression analyses, two representative candidates (HSP70 and POX18) were reconfirmed by additional yeast mating assays to further ensure the specificity of these interactions. A statement mentioning this limitation (lines 555-557) was also added in “Concluding Remarks”.

Query: Line 478-485 (now 495-503)- Confused with this paragraph:

Figure 4 and 5 show transcripts do change in abundance over time with growth, and the results also say that both Pox18 and Hsp70 exhibit "significant temporal modulation”.

Statement made on line 481 contrasts with sentence above - do you mean “regulated through post-translational mechanisms”?

Again, the sentence beginning on line 483 seems to contrast to the patterns seen in the figures.

Response: We thank the reviewer for this observation. The text discussing the expression dynamics of HSP70 and POX18 has been revised for clarity and consistency with the data. On lines 495-503, we now state that both genes exhibit moderate but significant temporal modulation in transcript abundance, mention their maximum levels of expression, and discuss that the magnitude of these changes suggests that regulation likely occurs primarily at the post-translational level.

Query: Line 486 (now 505)- From looking at the graph for CHX, although the transcript expression is only significantly different in Hsp70 control vs CHX treatment, the other transcripts show similar patterns (higher expression after CHX treatment vs control). I don’t think from these findings you can conclude that Hsp70 is under feedback transcript stabilisation versus SBiP1 or POX18 are transcriptionally stable and not linked to active translation. Maybe if more replicates were included all the genes would show the same significance. Authors could reword to say “Hsp70 shows a weak significant increase” and mention that the other transcripts show exactly the same pattern, but it doesn’t meet significance.

Response: We appreciate this constructive comment. The section has been modified on lines 505-507, to state that HSP70 shows a weak but significant increase in transcript levels, while SBiP1 and POX18 follow the same upward trend without reaching statistical significance. The part of a “feedback stabilization” mechanism was removed, and the paragraph now indicates that translation inhibition may cause a general, modest accumulation of transcripts rather than a gene-specific regulatory effect.

Query: Line 503 (now 520)- not sure if the findings confirm that SBiP1 is a “central hub” due to Y2H assay limitations - reword as the authors need extra confirmation of protein-protein interactions before making this statement

Response: We have modified this statement accordingly and now only suggest an initial proposed role as regulatory hub for SBiP1 (lines 520-521), and on line 552-553, we now state that “our study suggests that SBiP1 may function as a potential regulatory hub in the ER homeostasis network” rather than as a confirmed central hub. We also acknowledge that additional experimental validation beyond Y2H is required to confirm these interactions and establish a SBiP1 integrative role.

Query: Typo in Figure 5 - should say “Hsp70” in graph title not “Hxp70”.

Response: The typo in Figure 5 has been corrected (“Hsp70” instead of “Hxp70”).

Query: Line 382 (now 391)- the text says that the expression of Hsp70 peaked at day 12 but figure 5 shows it peaking at day 6. The numbers in the text for change in expression on day 12 and 6 also don’t match those on the figure - looks to be around 6.5-fold for day 6 and 6.25-fold for day 12.

Response: We appreciate the reviewer’s observation. The text in the Results section has been revised (lines 391-393) to accurately reflect the data shown in Figure 5 (also same response to second query above), now indicating that HSP70 expression increased sharply at day 6 and remained elevated at day 12 before decreasing toward baseline levels at later time points.

Reviewer #2: The manuscript is generally well written; however, there are several areas where improvements are needed.

Query: In the Abstract, acronyms should be defined before they are used. For example, ER appears without being defined anywhere in the manuscript.

Response: We thank the reviewer for noting this oversight. The acronym “ER” has now been defined as “endoplasmic reticulum (ER)” on line 67 in the Introduction section and checked for consistency throughout the manuscript.

Query: Additionally, the figures could be improved by enhancing their resolution, increasing clarity of labels, and ensuring that all elements are consistently formatted and redundant info removed (e.g., parts in brackets for Figures 2-6).

Response: We appreciate the reviewer’s feedback. All figures (1–6) were reformatted to improve resolution and label clarity, ensuring consistent font size, color palette, and axis style. Redundant elements in figure titles were removed for a cleaner and more uniform presentation.

Query: Can author add limitation regarding the way how gene expression was assessed as using only one reference genes may not be sufficient. Why was this specific reference gene SmicEF1-α used? Did you consider other genes? What was validation process here?

Response: We added in the text extensive background to support the use of EF1-alpha as the reference gene above any others due to its adequate performance in qPCR assays in numerous instances of species, developmental stages and culture conditions including microalgae and Symbiodiniaceae (references 35-42). The mentioned corresponding references have been added to the text in the Methods (lines 226-227), Results (line 323), and Discussion (536-537) sections. Furthermore, we have also provided additional support for the validity of the data on lines 490-492 and 540-544, by mentioning our previous results using western immunoblotting [reference 40], where we found constant SBiP1 protein expression under either dark or light conditions and which are fully coincident with part of the results from this study (Fig. 3).

---

## [Decision Letter · Decision Letter 1]

16 Dec 2025

Dear Dr. Villanueva,

Please address the remaining concern of Reviewer 1 in a revised manuscript. Please submit your revised manuscript by Jan 30 2026 11:59PM. If you will need more time than this to complete your revisions, please reply to this message or contact the journal office at plosone@plos.org . A letter that responds to each point raised by the academic editor and reviewer(s). You should upload this letter as a separate file labeled 'Response to Reviewers'.A marked-up copy of your manuscript that highlights changes made to the original version. You should upload this as a separate file labeled 'Revised Manuscript with Track Changes'.An unmarked version of your revised paper without tracked changes. You should upload this as a separate file labeled 'Manuscript'.

We look forward to receiving your revised manuscript.

Kind regards,

Katherine A. Borkovich, Ph.D.

Academic Editor

PLOS One

Journal Requirements:

Reviewers' comments:

Reviewer's Responses to Questions

**Comments to the Author**

Reviewer #1: (No Response)

2. Is the manuscript technically sound, and do the data support the conclusions?

Reviewer #1: Partly

3. Has the statistical analysis been performed appropriately and rigorously?

Reviewer #1: Yes

4. Have the authors made all data underlying the findings in their manuscript fully available?

Reviewer #1: Yes

5. Is the manuscript presented in an intelligible fashion and written in standard English?

Reviewer #1: Yes

Reviewer #1: The authors have nicely addressed the points detailed in the previous review round. I have one more comment after re-reading the manuscript that would be nice for the authors to elaborate on and clarify their conclusions.

Paragraph starting line 389 and the sentences 498-503. The authors say that the “magnitude of the fluctuations” of the changes in gene expression in Hsp70 and Pox18 implies that they may be regulated at the post-translational level. I wasn’t too sure whether the authors are implying the change in gene expression is low? However, after looking at the graphs, the change in expression seems quite high for Symbiodiniaceae (up to 4.21 fold for Pox18 and 6.32 fold for Hsp70) versus other papers that say most genes show minimal changes in expression (e.g., Gierz et al. 2017) and that’s why other papers have hypothesised that most regulation may be done at the post-translational level. From the data presented here in this current study, it seems like the opposite, where Hsp70 and Pox18 may have high regulation at the transcript level. The authors do say this in lines 393-395 but the sentence following (line 395-398) contradicts this statement. Please could the authors clarify this.

**Do you want your identity to be public for this peer review?** For information about this choice, including consent withdrawal, please see our Privacy Policy

Reviewer #1: **Yes:**  Lucy Gorman

---

## [Author Response · Author response to Decision Letter 2]

17 Dec 2025

Response to Reviewer #1.

Reviewer #1: The authors have nicely addressed the points detailed in the previous review round. I have one more comment after re-reading the manuscript that would be nice for the authors to elaborate on and clarify their conclusions.

Paragraph starting line 389 and the sentences 498-503. The authors say that the “magnitude of the fluctuations” of the changes in gene expression in Hsp70 and Pox18 implies that they may be regulated at the post-translational level. I wasn’t too sure whether the authors are implying the change in gene expression is low? However, after looking at the graphs, the change in expression seems quite high for Symbiodiniaceae (up to 4.21 fold for Pox18 and 6.32 fold for Hsp70) versus other papers that say most genes show minimal changes in expression (e.g., Gierz et al. 2017) and that’s why other papers have hypothesised that most regulation may be done at the post-translational level. From the data presented here in this current study, it seems like the opposite, where Hsp70 and Pox18 may have high regulation at the transcript level. The authors do say this in lines 393-395 but the sentence following (line 395-398) contradicts this statement. Please could the authors clarify this.

Response:

Thank you for raising this observation. We agree that the observed fold-changes (up to ~6.3× for HSP70 and ~4.2× for POX18) in expression are relatively high compared to most Symbiodiniaceae genes, which typically show minor variation in transcript levels.

To address this query, we have revised the text in both the Results (lines 396-401) and Discussion (lines 501-508) sections. We have added that previous reports document low-level transcriptional changes in Symbiodiniaceae as noted by the reviewer (Gierz et al. 2017; reference [3]) and that our observed data suggest a transcriptional component in the regulation of HSP70 and POX18, while post-translational mechanisms likely act also in parallel to fine-tune functional regulation.

---

## [Editor Report · Decision Letter 2]

21 Dec 2025

Mapping SBiP1 protein-protein interactions in Symbiodinium microadriaticum CassKB8 using the Yeast Two-Hybrid assay and structural prediction

PONE-D-25-54690R2

Dear Dr. Villanueva,

We’re pleased to inform you that your manuscript has been judged scientifically suitable for publication and will be formally accepted for publication once it meets all outstanding technical requirements.

Kind regards,

Katherine A. Borkovich, Ph.D.

Academic Editor

PLOS One
---

## [Editor Report · Acceptance letter]

PONE-D-25-54690R2

PLOS One

Dear Dr. Villanueva,

I'm pleased to inform you that your manuscript has been deemed suitable for publication in PLOS One. Congratulations! Your manuscript is now being handed over to our production team.

Kind regards,

on behalf of

Dr. Katherine A. Borkovich

Academic Editor

PLOS One